# A New Class of 2$q$-Point Nonstationary Subdivision Schemes and Their Applications

**Abdul Ghaffar** [1,†], **Mehwish Bari** [2,†], **Zafar Ullah** [3,†], **Mudassar Iqbal** [1,†], **Kottakkaran Sooppy Nisar** [4,*,†] **and and Dumitru Baleanu** [5,6,†]

1   Department of Mathematical Sciences, BUITEMS, Quetta 87300, Pakistan
2   Department of Mathematics, NCBA&E, Bahawalpur 63100, Pakistan
3   Department of Mathematics, University of Education Lahore, Campus DG Khan,
    Dera Ghazi Khan 54770, Pakistan
4   Department of Mathematics, College of Arts and Sciences, Prince Sattam bin Abdulaziz University,
    Wadi Aldawaser 11991, Saudi Arabia
5   Department of Mathematics, Cankaya University, Ankara 06530, Turkey
6   Institute of Space Sciences, Magurele-Bucharest 76900, Romania
*   Correspondence: n.sooppy@psau.edu.sa; Tel.: +966-563456976
†   These authors contributed equally to this work.

**Abstract:** The main objective of this study is to introduce a new class of 2$q$-point approximating nonstationary subdivision schemes (ANSSs) by applying Lagrange-like interpolant. The theory of asymptotic equivalence is applied to find the continuity of the ANSSs. These schemes can be nicely generalized to contain local shape parameters that allow the user to locally adjust the shape of the limit curve/surface. Moreover, many existing approximating stationary subdivision schemes (ASSSs) can be obtained as nonstationary counterparts of the proposed ANSSs.

**Keywords:** stationary; nonstationary; subdivision scheme; continuity; curvature and torsion

**MSC:** 65D17, 65D07, 65D05

## 1. Introduction

The importance of subdivision schemes (SSs) cannot be denied because it plays an important role in computer aided geometric design (CAGD), geometric modeling, computer graphics, image processing, visualization and engineering, etc. Nowadays, SSs can be distinguished in various types: Range from uniform to non-uniform, binary to an arbitrary arity, interpolatory to approximating, and stationary to nonstationary. It seems that stationary subdivision schemes (SSSs) have interesting features, but reconstruction of special types of limit curves of various shapes—including polynomial functions and conic sections such as circles, ellipses, and spiral curves—could not be accomplished without the nonstationary subdivision schemes (NSSs). First of all, De Rham [1] constructed approximating SSS and, later on, Chaikin [2] introduced the corners cutting SSS. Further, Deslauriers and Dubuc [3] established a four-point interpolating SSS.

In literature, several articles have been published during the last couple of decades in the field of SSs. The initiative research in the case of univariate SSs were limited to SSSs. In 1991, Dyn and Levin [4] introduced the 1st NSS. In 2003, Jena et al. [5] constructed a four-point interpolating NSS generating limit curves of $C^1$-continuity. In 2005, Curve subdivision schemes on manifolds and in Lie groups [6] were constructed from linear subdivision schemes by first representing the rules of affinely invariant linear schemes in terms of repeated affine averages, and then replacing the operation of affine average either by a geodesic average, or by projection of the affine averages

onto a surface. In 2007, Beccari et al. [7] derived a nonstationary binary four-point uniform tension controlled interpolating SS reproducing conics and they also proposed another ternary four-point $C^2$-continuous interpolating NSS with tension control [8] in the same year. In 2007, Daniel and Shunmugaraj [9] introduced some four-point ternary interpolating nonstationary schemes. In 2009, Daniel and Shunmugaraj [10] introduced six-point binary interpolating NSS that is $C^2$ limit curve. Conti and Romani [11] presented and investigated a six-point interpolatory NSS capable of reproducing important curves. In 2009, Dyn et al. [12] investigated a four-point curve SS based on iterated chordal and centripetal parameterizations. In 2013, Li et al. [13] developed a new technique to establish NSS that can generate functions in a finite-dimensional subspace of exponential polynomials. In 2014, Siddiqi and Younis [14] introduced an algorithm to produce a family of ternary ANSSs for curve design. In 2015, Conti et al. [15] established a new equivalence notion between NSSs, termed asymptotic similarity, which is weaker than asymptotic equivalence. In 2016, Salam et al. [16] presented two nonstationary forms of Chaikin perturbation SS, and Tan et al. [17] derived a three-point ANSS. In 2017, Hameed and Mustafa [18] presented a generalized Refine–Smooth algorithm to design a family of $a$-point $b$-ary approximating SS with a bell-shaped mask, where $a \geq 3$ and $b \geq 2$. In 2018, Zhang et al. [19] proposed a new combined approximating and interpolating ternary four-point SS with multiple parameters. For more details about the generalizations of SSSs, readers can refer to [18,20–28].

This motivated us to present a new class of binary ANSSs with high smoothness and more degrees of freedom for the curve design. The proposed schemes not only provide the mask of even-point NSSs, but also generalize and unify several well-known SSs. In order to show the performance of the these schemes, we compare continuity, shape of limit curves, curvature, and torsion numerically. Sufficient conditions on the shape control parameter $u_0$ are developed that allow us to generate curvature and torsion after a finite number of subdivision steps. Moreover, the limit curves with specific value of shape control parameter $u_0$ are depicted by significant application of derived conditions on the initial data. It is observed that the limit curves of our approximating schemes are near the initial control polygons and—for a certain range of parameters—limit curves pass through the initial polygons.

The paper is organized as follows: In Section 1, we present preliminaries and establish some lemmas. In Section 2, we offer an algorithm to produce a new family of $2q$-point binary ANSSs. Smoothness and convergence of the $2q$-point binary ANSSs are discussed in Section 3. The results and discussion are given in Section 4, while Section 5 is devoted to the conclusion.

## 2. Preliminaries

A general form of univariate binary subdivision scheme $S$ which maps a polygon $t^j = \{t_i^j\}_{i \in \mathbb{Z}}$ to a refined polygon $t^{j+1} = \{t_i^{j+1}\}_{i \in \mathbb{Z}}$ is defined by

$$t_{2i+l}^{j+1} = \sum_{k=-(q-1)}^{q} a_{2i+l}^j t_{i+j}^j, \quad l = 0, 1, \tag{1}$$

where $q > 0$ and $\mathbb{Z}$ are the set of integers. The set of coefficients $\{a_{2i+l}^j, \; l = 0, 1\}$ is called the subdivision mask. This scheme (1) is formally denoted by $t^{j+1} = S_j t^j$.
A necessary condition for uniform convergence of the subdivision scheme (1) is

$$\sum_{i \in \mathbb{Z}} a_{2i}^j = \sum_{i \in \mathbb{Z}} a_{2i+1}^j = 1. \tag{2}$$

If the mask $a^j$ is independent of $j$, the subdivision scheme $S_{a^j}$ corresponding to the mask $a^j$ is called stationary; otherwise it is called nonstationary.

**Theorem 1.** *[29] Two binary SSs, $S_a^j$ and $S_b^j$, are asymptotically equivalent if*

$$\sum_{j=1}^{\infty} \left\| S_a^j - S_b^j \right\| < \infty,$$

*where*

$$\|S_{a^j}\| = max \left\{ \sum_{i \in z} \left| a_{2i}^j \right|, \sum_{i \in z} \left| a_{2i+1}^j \right| \right\}.$$

**Theorem 2.** *[29] Let $S_a^j$ and $S_a$ be binary nonstationary and stationary SSs, respectively, which are said to be asymptotically equivalent if they have finite masks of the same support. If stationary scheme $S_a$ is $C^m$ and*

$$\sum_{j=0}^{\infty} 2^{mj} \|S_a^j - S_a\|_\infty < \infty,$$

*then the nonstationary scheme $S_a^j$ is also $C^m$.*

Construction of SSs using Lagrange interpolation was presented by Deslauriers and Dubuc [21]. We also use the Lagrange polynomial to construct a class of nonstationary SSs.

Consider the Lagrange polynomial of degree $(2q - 1)$, for any integer $q \leq 2$, corresponding to node $\{p\}_{-(q-1)}^q$,

$$L_p^{2q-1}(y) = \prod_{k=-(q-1), p \neq k}^{q} \frac{y - k}{p - k}, \qquad p = -(q-1), -(q-2), \cdots, (q). \tag{3}$$

First, we present some preliminary identities which play an important role in this Section.

**Lemma 1.** *If $p = -(q-1), ..., (q)$ and $q$ is integer, then the following implication holds:*

$$A_p := \prod_{k=-(q-1), \ k \neq p}^{q} (p - k) = (-1)^{q-p}(q + p - 1)!(q - p)!. \tag{4}$$

**Proof.** Now, we prove the above result for each value of $p$. Here, for $p = -(q-1)$, we get

$$\prod_{k=-(q-1), k \neq p}^{q} (p - k) = (-1)(-2)(-3)\ldots(-2q + 2)(-2q + 1).$$

$$\Rightarrow$$

$$\prod_{k=-(q-1), k \neq p}^{q} (p - k) = 0!(-1)^{2q-1}(2q - 1)!.$$

Since $p = -(q-1)$, then the above equation can be expressed in the form of (4). Similarly for $p = -(q-2), \ldots, 0, \ldots, q$, we have (4). This completes the proof.  $\square$

**Lemma 2.** *If $L_p^{2q-1}(y)$ is a Lagrange fundamental polynomial of degree $(2q - 1)$ defined by (3), corresponding to the nodes $\{p\}_{-(q-1)}^q$, then we have*

$$B_p := L_p^{2q-1}\left(\frac{1}{4}\right) = \frac{(-1)^p(4q - 1)(4q - 3)!}{2^{6q-4}(1 - 4p)(2q - 2)!(q + p - 1)!(q - p)!}. \tag{5}$$

**Proof.** Since

$$\prod_{k=-(q-1)}^{q} \left( \frac{1}{4} - k \right) = \left( \frac{1}{4} \right)^{2q} \{ (4q-3)(4q-7)(4q-11) \cdots (5)(1)(-3) \cdots$$

$$(-4q+13)(-4q+9)(-4q+5)(-4q+1) \},$$

then

$$\prod_{k=-(q-1),k\neq k}^{q} \left( \frac{1}{4} - k \right) = \frac{1}{4^{2q-1}(1-4k)} \prod_{k=-q+1}^{q} (1-4k).$$

This implies

$$\prod_{k=-(q-1),k\neq p}^{q} \left( \frac{1}{4} - k \right) = \frac{(-1)^q}{4^{2q-1}(1-4p)} \left\{ (4q-3) \frac{(4q-4)}{(4q-4)} \frac{(4q-5)}{(4q-5)} \frac{(4q-6)}{(4q-6)} \right.$$

$$(4q-7) \frac{(4q-8)}{(4q-8)} \frac{(4q-9)}{(4q-9)} \frac{(4q-10)}{(4q-10)} (4q-11) \frac{(4q-12)}{(4q-12)}$$

$$\cdots \left( \frac{8}{8} \right) \left( \frac{7}{7} \right) \left( \frac{6}{6} \right) (5) \left( \frac{4}{4} \right) \left( \frac{3}{3} \right) \left( \frac{2}{2} \right) (1) \left( \frac{2}{2} \right) (3)$$

$$\left( \frac{4}{4} \right) \left( \frac{5}{5} \right) \left( \frac{6}{6} \right) (7) \dots (4q-13) \frac{(4q-12)}{(4q-12)} \frac{(4q-11)}{(4q-11)}$$

$$\frac{(4q-10)}{(4q-10)} (4q-9) \frac{(4q-8)}{(4q-8)} \frac{(4q-7)}{(4q-7)} \frac{(4q-6)}{(4q-6)}$$

$$\left. (4q-5) \frac{(4q-4)}{(4q-4)} \frac{(4q-3)}{(4q-3)} \frac{(4q-2)}{(4q-2)} (4q-1) \right\}.$$

This leads to

$$\prod_{k=-(q-1),k\neq p}^{q} \left( \frac{1}{4} - k \right) = \frac{(-1)^q (4q-3)!(4q-1)}{2^{6q-4}(1-4p)(2q-2)!}.$$

This completes the proof.  □

### 3. The Even-Point ANSSs

In this section, we present general explicit formulae to construct the mask of $2q$-point nonstationary binary approximating subdivision schemes.

If we construct a binary ANSS with

$$a_k(z) = (1+z)^{2q-1} \sum_{i=0}^{2q} \binom{2q}{i} u_i x^i, \quad k = -q+1, ..., q, \tag{6}$$

where

$$\sum_{i=0}^{2q} \binom{2q}{i} u_i = 0, \; u_k = u_{2q-k}, \, k = 0, 1, \dots, 2q-1. \tag{7}$$

From (6) and (7), we get new vertex position

$$t_{2i+\alpha}^{j+1} = \sum_{k=-(q-1)}^{q} a_k t_{i+k}^{j} \quad \alpha = 0, 1, \tag{8}$$

where $i \in \mathbb{Z}$ and $j \in \mathbb{Z}^+$.

For $q = 1$ in (6), we have

$$
\begin{aligned}
a_k(z) \quad &= (1+z) \sum_{i=0}^{2} \binom{2}{i} u_i z^i \\
&= (1+z) \left( \binom{2}{0} u_0 z^0 + \binom{2}{1} u_1 z^1 + \binom{2}{2} u_2 z^2 \right) \\
&= (1+z)(u_0 + 2u_1 z + u_2 z^2).
\end{aligned}
\tag{9}
$$

For $q = 1$ in (7), we have

$$
\sum_{i=0}^{2} \binom{2}{i} u_i = 0
$$

This implies

$$
\binom{2}{0} u_0 + \binom{2}{1} u_1 + \binom{2}{2} u_2 = (u_0 + 2u_1 + u_2) = 0
\tag{10}
$$

and

$$
u_0 = u_2, \quad u_1 = u_1,
$$

putting in (10), we get $u_0 + 2u_1 + u_0 = 0$ or $u_1 = -u_0$.

Putting in (9), we have

$$
a_k(z) = (1+z)(u_0 - 2u_0 z + u_0 x^2) = (u_0 - u_0 z - u_0 x^2 + u_0 z^3).
\tag{11}
$$

From the above mask, we get

$$
t_{2i}^{j+1} = -u_0 t_i^j + u_0 t_{i+1}^j, \, t_{i+1}^j = u_0 t_i^j - u_0 t_{i+1}^j,
\tag{12}
$$

where $a_0 = u_0$ and $a_1 = -u_0$.

For $q = 2$ in (6), we have

$$
\begin{aligned}
a_k(z) \quad &= (1+z)^3 \sum_{i=0}^{4} \binom{4}{i} u_i z^i \\
&= (1+z)^3 \left( \binom{4}{0} u_0 z^0 + \binom{4}{1} u_1 z^1 + \binom{4}{2} u_2 z^2 + \binom{4}{3} u_3 z^3 + \binom{4}{4} u_4 z^4 \right) \\
&= (1+z)^3 (u_0 + 4u_1 z + 6u_2 z^2 + 4u_3 z^3 + u_4 z^4).
\end{aligned}
\tag{13}
$$

Applying the same process, we have

$$
\begin{aligned}
a_k(z) \quad &= (1+z)^3 (u_0 - 2u_0 z^2 + u_0 z^4) \\
&= (u_0 + u_0 z + 3u_0 z^2 - 5u_0 z^3 - 5u_0 z^4 + 3u_0 z^5 + u_0 z^6 + u_0 z^7).
\end{aligned}
\tag{14}
$$

From the above mask, we get

$$
\begin{cases}
t_{2i}^{j+1} = u_0 t_{i-1}^j - 5u_0 t_i^j + 3u_0 t_{i+1}^j + u_0 t_{i+2}^j, \\
t_{2i+1}^{j+1} = u_0 t_{i-1}^j + 3u_0 t_i^j - 5u_0 t_{i+1}^j + u_0 t_{i+2}^j.
\end{cases}
\tag{15}
$$

$$
\begin{cases}
t_{2i}^{j+1} = \sum_{k=-q}^{q-1} a_{-k}\, t_{i+k}^{j}, \\
t_{2i+1}^{j+1} = \sum_{k=-(q-1)}^{q} a_k\, t_{i+k}^{j},
\end{cases} \tag{16}
$$

where $a_{-1} = u_0$, $a_0 = 3u_0$, $a_1 = -5u_0$, and $a_2 = u_0$.

Given $q \geq 1$, the $2q$-point nonstationary approximating schemes are defined by

$$
\begin{cases}
t_{2i}^{j+1} = \sum_{k=-q}^{q-1} \lambda_{-k}^{j}\, t_{i+k}^{j}, \\
t_{2i+1}^{j+1} = \sum_{k=-(q-1)}^{q} \lambda_{k}^{j}\, t_{i+k}^{j},
\end{cases} \tag{17}
$$

where

$$
\lambda_k^j = \frac{\sin\left(\frac{\theta}{2^{j+1}} B_p\right)}{\sin\left(\frac{\theta}{2^{j+1}} A_p\right)} + \frac{\sin\left(\frac{\theta}{2^{j+1}} a_k\right)}{\sin\left(\frac{\theta}{2^{j+1}} 4^{2q-1}\right)}, \quad p = -q+1, \dots, q \ \& \ 0 < \theta < \frac{\pi}{2},
$$

$A_p$, $B_p$, and $a_k$ are defined by (5) and (8), respectively.

Examples:

- Using $q = 1$ in (17) and (18), we get a new 2-point symmetric binary ANSS with free parameter $u_0$

$$
\begin{aligned}
t_{2i}^{j+1} &= \lambda_0^j t_i^j + \lambda_1^j t_{i+1}^j, \\
t_{2i+1}^{j+1} &= \lambda_1^j t_i^j + \lambda_0^j t_{i+1}^j,
\end{aligned} \tag{18}
$$

where

$$
\lambda_0^j = \frac{\sin\left(\frac{3\theta}{2^{j+1}}\right) - \sin\left(\frac{\theta u_0}{2^{j+1}}\right)}{\sin\left(\frac{\theta}{2^{j-1}}\right)},
$$

$$
\lambda_1^j = \frac{\sin\left(\frac{\theta}{2^{j+1}}\right) + \sin\left(\frac{\theta u_0}{2^{j+1}}\right)}{\sin\left(\frac{\theta}{2^{j-1}}\right)}.
$$

In fact, the sum of the stencils of scheme at the $j$th level are

$$
\lambda^j = \sum_{p=-q+1}^{q} \lambda_p^j.
$$

Now for $q = 1$, we define the normalized ANSS (corresponding to (18)). Observe that

$$
\begin{aligned}
\lambda^j &= \lambda_0^j + \lambda_1^j = \frac{\sin\left(\frac{3\theta}{2^{j+1}}\right) - \sin\left(\frac{\theta u_0}{2^{j+1}}\right)}{\sin\left(\frac{\theta}{2^{j-1}}\right)} + \frac{\sin\left(\frac{\theta}{2^{j+1}}\right) + \sin\left(\frac{\theta u_0}{2^{j+1}}\right)}{\sin\left(\frac{\theta}{2^{j-1}}\right)} \\
&= \frac{1}{\sin\left(\frac{\theta}{2^{j-1}}\right)} \left(\sin\left(\frac{3\theta}{2^{j+1}}\right) + \sin\left(\frac{\theta}{2^{j+1}}\right)\right) \\
&= \frac{1}{\sin\left(\frac{\theta}{2^{j-1}}\right)} \left(2 \sin\left(\frac{2\theta}{2^{j+1}}\right) \cos\left(\frac{\theta}{2^{j+1}}\right)\right) = \frac{\cos\left(\frac{\theta}{2^{j+1}}\right)}{\cos\left(\frac{\theta}{2^j}\right)}.
\end{aligned}
$$

The corresponding normalized SS can be obtained by dividing the stencils at the $j$th iteration of SS (18) by their sum.

$$
\begin{aligned}
t_{2i}^{j+1} &= \mu_0^j t_i^j + \mu_1^j t_{i+1}^j, \\
t_{2i+1}^{j+1} &= \mu_1^j t_i^j + \mu_0^j t_{i+1}^j,
\end{aligned}
\tag{19}
$$

where

$$
\mu_0^j = \frac{\cos\left(\frac{\theta}{2^j}\right)}{\cos\left(\frac{\theta}{2^{j+1}}\right)} \lambda_0^j, \quad
\mu_1^j = \frac{\cos\left(\frac{\theta}{2^j}\right)}{\cos\left(\frac{\theta}{2^{j+1}}\right)} \lambda_1^j.
$$

**Lemma 3.** *If $f$ is the limit function of ANSS (18), then $(\cos\theta)f(y)$ is the limit function of the corresponding normalized SS.*

**Proof.** Note that

$$
\begin{aligned}
\lim_{q\to\infty} \prod_{j=0}^{q} \frac{1}{\lambda_0^j + \lambda_1^j}
&= \lim_{q\to\infty} \prod_{j=0}^{q} \frac{\cos\left(\frac{\theta}{2^j}\right)}{\cos\left(\frac{\theta}{2^{j+1}}\right)} \\
&= \lim_{q\to\infty} \frac{\cos\theta}{\cos\left(\frac{\theta}{2^{q+1}}\right)} \\
&= \cos\theta.
\end{aligned}
$$

□

- Using $q = 2$ in (17) and (18), we have a new four-point symmetric ANSS with free parameter $u_0$

$$
\begin{aligned}
t_{2i}^{j+1} &= \lambda_{-1}^j t_{i-1}^j + \lambda_0^j t_i^j + \lambda_1^j t_{i+1}^j + \lambda_2^j t_{i+2}^j, \\
t_{2i+1}^{j+1} &= \lambda_2^j t_{i-1}^j + \lambda_1^j t_i^j + \lambda_0^j t_{i+1}^j + \lambda_{-1}^j t_{i+2}^j,
\end{aligned}
\tag{20}
$$

where

$$
\lambda_{-1}^j = \frac{\sin\left(-\frac{7\theta}{2^{j+1}}\right)}{\sin\left(\frac{32\theta}{2^{j-1}}\right)} + \frac{\sin\left(\frac{3u_0\theta}{2^{j+1}}\right)}{\sin\left(\frac{64\theta}{2^{j+1}}\right)},
$$

$$
\lambda_0^j = \frac{\sin\left(\frac{105\theta}{2^{j+1}}\right)}{\sin\left(\frac{32\theta}{2^{j-1}}\right)} - \frac{\sin\left(\frac{5u_0\theta}{2^{j+1}}\right)}{\sin\left(\frac{64\theta}{2^{j+1}}\right)},
$$

$$
\lambda_1^j = \frac{\sin\left(\frac{35\theta}{2^{j+1}}\right)}{\sin\left(\frac{32\theta}{2^{j-1}}\right)} + \frac{\sin\left(\frac{u_0\theta}{2^{j+1}}\right)}{\sin\left(\frac{64\theta}{2^{j+1}}\right)},
$$

$$
\lambda_2^j = \frac{\sin\left(-\frac{5\theta}{2^{j+1}}\right)}{\sin\left(\frac{32\theta}{2^{j-1}}\right)} + \frac{\sin\left(\frac{u_0\theta}{2^{j+1}}\right)}{\sin\left(\frac{64\theta}{2^{j+1}}\right)}.
$$

Now, substituting $q = 2$, we define the normalized ANSS (corresponding to (21)). Observe that

$$
\begin{aligned}
\lambda^j &= \lambda^j_{-1} + \lambda^j_0 + \lambda^j_1 + \lambda^j_2 \\
&= \frac{\sin\left(-\frac{7\theta}{2^{j+1}}\right)}{\sin\left(\frac{128\theta}{2^{j+1}}\right)} + \frac{\sin\left(\frac{3u_0\theta}{2^{j+1}}\right)}{\sin\left(\frac{64\theta}{2^{j+1}}\right)} + \frac{\sin\left(\frac{105\theta}{2^{j+1}}\right)}{\sin\left(\frac{128\theta}{2^{j+1}}\right)} - \frac{\sin\left(\frac{5u_0\theta}{2^{j+1}}\right)}{\sin\left(\frac{64\theta}{2^{j+1}}\right)} + \frac{\sin\left(\frac{35\theta}{2^{j+1}}\right)}{\sin\left(\frac{128\theta}{2^{j+1}}\right)} \\
&\quad + \frac{\sin\left(\frac{u_0\theta}{2^{j+1}}\right)}{\sin\left(\frac{64\theta}{2^{j+1}}\right)} + \frac{\sin\left(\frac{-5\theta}{2^{j+1}}\right)}{\sin\left(\frac{128\theta}{2^{j+1}}\right)} + \frac{\sin\left(\frac{u_0\theta}{2^{j+1}}\right)}{\sin\left(\frac{64\theta}{2^{j+1}}\right)}.
\end{aligned}
$$

Similarly, the corresponding normalized scheme can be obtained by dividing the stencils at the $j$th level of ANSS (20) by their sum.

$$
\begin{aligned}
t^{j+1}_{2i} &= \mu^j_{-1} t^j_{i-1} + \mu^j_0 t^j_i + \mu^j_1 t^j_{i+1} + \mu^j_2 t^j_{i+2}, \\
t^{j+1}_{2i+1} &= \mu^j_2 t^j_{i-1} + \mu^j_1 t^j_i + \mu^j_0 t^j_{i+1} + \mu^j_{-1} t^j_{i+2},
\end{aligned}
\tag{21}
$$

where

$$
\mu^j_{-1} = \frac{\frac{\sin\left(-\frac{7\theta}{2^{j+1}}\right)}{\sin\left(\frac{32\theta}{2^{j-1}}\right)} + \frac{\sin\left(\frac{3u_0\theta}{2^{j+1}}\right)}{\sin\left(\frac{64\theta}{2^{j+1}}\right)}}{\lambda^j},
$$

$$
\mu^j_0 = \frac{\frac{\sin\left(\frac{105\theta}{2^{j+1}}\right)}{\sin\left(\frac{32\theta}{2^{j-1}}\right)} - \frac{\sin\left(\frac{5u_0\theta}{2^{j+1}}\right)}{\sin\left(\frac{64\theta}{2^{j+1}}\right)}}{\lambda^j},
$$

$$
\mu^j_1 = \frac{\frac{\sin\left(\frac{35\theta}{2^{j+1}}\right)}{\sin\left(\frac{32\theta}{2^{j-1}}\right)} + \frac{\sin\left(\frac{u_0\theta}{2^{j+1}}\right)}{\sin\left(\frac{64\theta}{2^{j+1}}\right)}}{\lambda^j},
$$

$$
\mu^j_2 = \frac{\frac{\sin\left(-\frac{5\theta}{2^{j+1}}\right)}{\sin\left(\frac{32\theta}{2^{j-1}}\right)} + \frac{\sin\left(\frac{u_0\theta}{2^{j+1}}\right)}{\sin\left(\frac{64\theta}{2^{j+1}}\right)}}{\lambda^j}.
$$

The normalized scheme (19) generates the function $f(y) = 1$ because $\sum \mu^j_p = 1$, $p = -q + 1, ..., q$. Interestingly, the proposed schemes (19) and (21) reproduce circles and can be demonstrated by the following lemma.

**Lemma 4.** *Let $j \geq 0$ & $q > 0$ be the integers. Let $t^j_i = \cos\left((2i)\frac{\theta}{2^j}\right)$, then, for $-1 \leq i \leq 2^j q$*

$$
t^{j+1}_{2i} = \cos\left(\left(2i + \frac{1}{2}\right)\frac{\theta}{2^j}\right) \quad \& \quad t^{j+1}_{2i+1} = \cos\left(\left(2i + \frac{3}{2}\right)\frac{\theta}{2^j}\right).
$$

*Similarly, if $t^j_i = \sin\left((2i)\frac{\theta}{2^j}\right)$, then, for $-1 \leq i \leq 2^j q$*

$$
t^{j+1}_{2i} = \sin\left(\left(2i + \frac{1}{2}\right)\frac{\theta}{2^j}\right) \quad \& \quad t^{j+1}_{2i+1} = \sin\left(\left(2i + \frac{3}{2}\right)\frac{\theta}{2^j}\right).
$$

**Proof.** We give the proof of the 1st part; the proof of the 2nd part can be achieved by the similar method. Consider $t_i^0 = \cos(2i\theta)$. At the 1st level of the scheme (18), we get

$$
\begin{aligned}
t_{2i}^1 &= \lambda_0^0 \cos(2i\theta) + \lambda_1^0 \cos((2i+2)\theta) \\
&= \frac{\sin\left(\frac{3\theta}{2}\right) - \sin\left(\frac{\theta u_0}{2}\right)}{\sin(2\theta)} \cos(2i\theta) + \frac{\sin\left(\frac{\theta}{2}\right) + \sin\left(\frac{\theta u_0}{2}\right)}{\sin(2\theta)} \cos((2i+2)\theta).
\end{aligned}
$$

(22)

Using $u_0 = 0$ in (22), we get

$$
\begin{aligned}
t_{2i}^1 &= \frac{\sin\left(\frac{3\theta}{2}\right)}{\sin(2\theta)} \cos(2i\theta) + \frac{\sin\left(\frac{\theta}{2}\right)}{\sin(2\theta)} \cos((2i+2)\theta). \\
&= \frac{\sin\left(2\theta - \frac{\theta}{2}\right)}{\sin(2\theta)} \cos(2i\theta) + \frac{\sin\left(\frac{\theta}{2}\right)}{\sin(2\theta)} \cos((2i+2)\theta). \\
&= \cos\left(\frac{\theta}{2}\right) \cos(2i\theta) - \sin\left(\frac{\theta}{2}\right) \sin(2i\theta) \\
&= \cos\left(\left(2i + \frac{1}{2}\right)\theta\right).
\end{aligned}
$$

At the $j$th level of the scheme, we get

$$
\begin{aligned}
t_{2i}^{j+1} &= \lambda_0^j \cos\left(2i\frac{\theta}{2^j}\right) + \lambda_1^j \cos\left((2i+2)\frac{\theta}{2^j}\right) \\
&= \frac{\sin\left(\frac{3\theta}{2^{j+1}}\right) - \sin\left(\frac{\theta u_0}{2^{j+1}}\right)}{\sin\left(\frac{\theta}{2^{j-1}}\right)} \cos\left((2i)\frac{\theta}{2^j}\right) \\
&\quad + \frac{\sin\left(\frac{\theta}{2^{j+1}}\right) + \sin\left(\frac{\theta u_0}{2^{j+1}}\right)}{\sin\left(\frac{\theta}{2^{j-1}}\right)} \cos\left((2i+2)\frac{\theta}{2^j}\right).
\end{aligned}
$$

(23)

Using $u_0 = 0$ in (23), we get

$$
\begin{aligned}
t_{2i}^{j+1} &= \frac{\sin\left(\frac{3\theta}{2^{j+1}}\right)}{\sin\left(\frac{\theta}{2^{j-1}}\right)} \cos\left((2i)\frac{\theta}{2^j}\right) + \frac{\sin\left(\frac{\theta}{2^{j+1}}\right)}{\sin\left(\frac{\theta}{2^{j-1}}\right)} \cos\left((2i+2)\frac{\theta}{2^j}\right). \\
&= \frac{\sin\left(\frac{\theta}{2^{j-1}} - \frac{\theta}{2^{j+1}}\right)}{\sin\left(\frac{\theta}{2^{j-1}}\right)} \cos\left(2i\frac{\theta}{2^j}\right) + \frac{\sin\left(\frac{\theta}{2^{j+1}}\right)}{\sin\left(\frac{\theta}{2^{j-1}}\right)} \cos\left((2i+2)\frac{\theta}{2^j}\right). \\
&= \cos\left(\frac{\theta}{2^{j+1}}\right) \cos\left(2i\frac{\theta}{2^j}\right) - \sin\left(\frac{\theta}{2^{j+1}}\right) \sin\left(2i\frac{\theta}{2^j}\right) \\
&= \cos\left(\left(2i + \frac{1}{2}\right)\frac{\theta}{2^j}\right).
\end{aligned}
$$

Similarly, we can obtain the proof of the other part

$$
t_{2i+1}^j = \cos\left(\left(2i + \frac{3}{2}\right)\frac{\theta}{2^j}\right).
$$

Likewise, we can prove that the scheme (20) also generates the functions $\cos(\theta y)$ and $\sin(\theta y)$. $\square$

## 4. Smoothness and Convergence of ANSSs

The convergence and smoothness of SSs (19) and (21) have been investigated by applying the theory of asymptotic equivalence [29]. Let $S_{a^j}$ represent the scheme (19). For convenience, we will show $a_k^j(\theta)$ by $a_k^j$, and $k = -(q-1), ..., 0, ..., (q)$. The following lemmas will be utilized to find the smoothness of scheme (19).

**Lemma 5.** *For some $j \geq 0$ and $0 < \theta < \frac{\pi}{2}$.*

$$(a) \quad \frac{1}{4}(1 + u_0) \leq \mu_1^j \leq \frac{1}{4}(1 + u_0)\frac{1}{\cos\left(\frac{\theta}{2^{j-1}}\right)}$$

$$(b) \quad \frac{1}{4}(3 - u_0) \leq \mu_0^j \leq \frac{1}{4}(3 - u_0)\frac{1}{\cos\left(\frac{\theta}{2^{j-1}}\right)}.$$

**Proof.** We give the proof of the 1st part; the proof of the 2nd part can be achieved analogously.

$$
\begin{aligned}
\mu_1^j &= \frac{\cos\left(\frac{\theta}{2^j}\right)\left(\sin\left(\frac{\theta}{2^{j+1}}\right) + \sin\left(\frac{\theta u_0}{2^{j+1}}\right)\right)}{\cos\left(\frac{\theta}{2^{j+1}}\right)\sin\left(\frac{\theta}{2^{j-1}}\right)} \\
&= \frac{\cos\left(\frac{\theta}{2^j}\right)\sin\left(\frac{\theta}{2^{j+1}}\right)}{\cos\left(\frac{\theta}{2^{j+1}}\right)\sin\left(\frac{\theta}{2^{j-1}}\right)} + \frac{\cos\left(\frac{\theta}{2^j}\right)\sin\left(\frac{\theta u_0}{2^{j+1}}\right)}{\cos\left(\frac{\theta}{2^{j+1}}\right)\sin\left(\frac{\theta}{2^{j-1}}\right)} \\
&\geq \frac{\frac{\theta}{2^{j+1}}}{\frac{\theta}{2^{j-1}}} + \frac{\frac{u_0\theta}{2^{j+1}}}{\frac{\theta}{2^{j-1}}} = \frac{\frac{\theta}{2^{j+1}}(1 + u_0)}{\frac{\theta}{2^{j-1}}} = \frac{1}{4}(1 + u_0).
\end{aligned}
$$

Also

$$
\begin{aligned}
\mu_1^j &= \frac{\cos\left(\frac{\theta}{2^j}\right)\left(\sin\left(\frac{\theta}{2^{j+1}}\right) + \sin\left(\frac{\theta u_0}{2^{j+1}}\right)\right)}{\cos\left(\frac{\theta}{2^{j+1}}\right)\sin\left(\frac{\theta}{2^{j-1}}\right)} \\
&= \frac{\cos\left(\frac{\theta}{2^j}\right)\sin\left(\frac{\theta}{2^{j+1}}\right)}{\cos\left(\frac{\theta}{2^{j+1}}\right)\sin\left(\frac{\theta}{2^{j-1}}\right)} + \frac{\cos\left(\frac{\theta}{2^j}\right)\sin\left(\frac{\theta u_0}{2^{j+1}}\right)}{\cos\left(\frac{\theta}{2^{j+1}}\right)\sin\left(\frac{\theta}{2^{j-1}}\right)} \\
&\leq \frac{\frac{\sin\left(\frac{\theta}{2^j}\right)}{\frac{\theta}{2^j}}\sin\left(\frac{\theta}{2^{j+1}}\right)}{\frac{\sin\left(\frac{\theta}{2^{j+1}}\right)}{\frac{\theta}{2^{j+1}}}\sin\left(\frac{\theta}{2^{j-1}}\right)} + \frac{\frac{\sin\left(\frac{\theta}{2^j}\right)}{\frac{\theta}{2^j}}\sin\left(\frac{\theta u_0}{2^{j+1}}\right)}{\frac{\sin\left(\frac{\theta}{2^{j+1}}\right)}{\frac{\theta}{2^{j+1}}}\sin\left(\frac{\theta}{2^{j-1}}\right)} \\
&\leq \frac{\sin\left(\frac{\theta}{2^j}\right)}{2\sin\left(\frac{\theta}{2^{j-1}}\right)} + \frac{\sin\left(\frac{\theta}{2^j}\right)\sin\left(\frac{\theta u_0}{2^{j+1}}\right)}{2\sin\left(\frac{\theta}{2^{j+1}}\right)\sin\left(\frac{\theta}{2^{j-1}}\right)} \\
&\leq \frac{\frac{\theta}{2^j}}{2\frac{\theta}{2^{j-1}}\cos\left(\frac{\theta}{2^{j-1}}\right)} + \frac{2\cos\left(\frac{\theta}{2^{j+1}}\right)\sin\left(\frac{\theta}{2^{j+1}}\right)\sin\left(\frac{\theta u_0}{2^{j+1}}\right)}{2\sin\left(\frac{\theta}{2^{j+1}}\right)\sin\left(\frac{\theta}{2^{j-1}}\right)} \\
&\leq \frac{(1 + u_0)\frac{\theta}{2^{j+1}}}{\frac{\theta}{2^{j-1}}\cos\left(\frac{\theta}{2^{j-1}}\right)} = \frac{1}{4}(1 + u_0)\frac{1}{\cos\left(\frac{\theta}{2^{j-1}}\right)}.
\end{aligned}
$$

This proves $(a)$; and the proof of $(b)$ is similar. $\square$

The preceding Lemma can be obtained by using Lemma 5.

**Lemma 6.** *For some constant $C_0$ and $C_1$, independent of $j$, we have*

$$(a) \quad \left| \mu_1^j - \frac{1}{4} \left( 1 + u_0 \right) \right| \leq C_0 \frac{1}{2^{2j}}.$$

$$(b) \quad \left| \mu_0^j - \frac{1}{4} \left( 3 - u_0 \right) \right| \leq C_1 \frac{1}{2^{2j}}.$$

**Proof.** Using $(a)$ of Lemma 5, we get

$$
\begin{aligned}
\left| \mu_1^j - \left( \frac{1 + u_0}{4} \right) \right| &\leq \left( \frac{1 + u_0}{4} \right) \left( \frac{1 - \cos\left( \frac{\theta}{2^{j-1}} \right)}{\cos\left( \frac{\theta}{2^{j-1}} \right)} \right) \\
&\leq 2 \frac{\frac{1}{4} \left( 1 + u_0 \right)}{\cos\left( \frac{\theta}{2^{j-1}} \right)} \sin^2 \left( \frac{\theta}{2^j} \right) \\
&\leq \frac{\frac{1}{2} \left( 1 + u_0 \right)}{\cos\left( \frac{\theta}{2^{j-1}} \right)} \frac{\theta^2}{2^{2j}} \leq C_0 \frac{1}{2^{2j}}.
\end{aligned}
$$

These complete proof $(a)$; and the proof of $(b)$ is similar. $\square$

**Remark 1.** *The following SS is a nonstationary counterpart of our proposed SS (19):*

$$
\begin{cases}
t_{2i}^{j+1} = \frac{1}{4} \left( 3 - u_0 \right) t_i^j + \frac{1}{4} \left( 1 + u_0 \right) t_{i+1}^j, \\
t_{2i+1}^{j+1} = \frac{1}{4} \left( 1 + u_0 \right) t_i^j + \frac{1}{4} \left( 3 - u_0 \right) t_{i+1}^j,
\end{cases}
\tag{24}
$$

*The stencils of the proposed scheme (19) converge to the masks of (24), $\mu_1^j \to \frac{1}{4} \left( 1 + u_0 \right)$ and $\mu_0^k \to \frac{1}{4} \left( 3 - u_0 \right)$, as $j \to \infty$. The proofs of these convergences follow from Lemma 6.*

- *Using $u_0 = 0$ in (19), we get a nonstationary counterpart of SS [2].*
- *Using $u_0 = 4\omega - \frac{1}{3}$ in (19), we get a nonstationary counterpart of the 2-point SS of [24].*

**Lemma 7.** *The Laurent polynomial $a(x)$ of the SS (24),*

$$a(x) = \frac{1}{4} \left\{ (1 + u_0) + (3 - u_0) x^1 + (3 - u_0) x^2 + (1 + u_0) x^3 \right\},$$

*and a scheme $S_a$ corresponding to the Laurent polynomial $a(x)$ is $C^0$ continuity for $-1 < u_0 < 1$ and $C^1$ continuity for $u_0 = 0$.*

**Proof.** Since

$$c(x) = \frac{2a(x)}{(1 + x)} = 2 \left\{ \frac{1}{4} \left( 1 + u_0 \right) + \frac{1}{2} \left( 1 - u_0 \right) x^1 + \left( 1 + u_0 \right) x^2 \right\}.$$

Observed that for $-1 < u_0 < 1$

$$\left\| \frac{1}{2} S_c \right\| = \frac{1}{2} \max \left\{ \sum_{i \in \mathbb{Z}} |c_{2i}|, \sum_{i \in \mathbb{Z}} |c_{2i+1}| \right\} = \max \left\{ \left| \frac{1 + u_0}{2} \right|, \left| \frac{1 - u_0}{2} \right| \right\} < 1.$$

Hence, by [30], the SS $S_a$ is $C^0$. In order to prove $C^1$ smoothness, we put $u_0 = 0$ in $c(x)$ and we get

$$c(x) = \frac{1}{2} \left( 1 + 2x + x^2 \right).$$

If

$$d(x) = \frac{2c(x)}{(1+x)} = (1+x),$$

then, $\left\| \frac{1}{2} S_d \right\| = \frac{1}{2} \max \left\{ \sum_{k \in \mathbb{Z}} |d_{2k}|, \sum_{k \in \mathbb{Z}} |d_{2k+1}| \right\} = \max \left\{ \frac{1}{2}, \frac{1}{2} \right\} < 1.$ Hence, by [30] (Corollary 4.11), the scheme $S_a$ is $C^1$ continuity. □

**Lemma 8.** *The Laurent polynomial $a^j(x)$ of the jth iteration of NSS (19) can be written as $a^j(x) = \left( \frac{1+x}{2} \right) b^j(x)$, where*

$$b^j(x) = 2 \left\{ \mu_1^j + \left( \mu_0^j - \mu_1^j \right) x + \mu_1^j x^2 \right\}.$$

**Proof.** Note that

$$a^j(x) = \mu_1^j + \left( \mu_0^j \right) x + \left( \mu_0^j \right) x^2 + \left( \mu_1^j \right) x^3.$$

Now, we can see that it is easily verified that $a^j(x) = \left( \frac{1+x}{2} \right) b^j(x).$ □

**Theorem 3.** *The NSSs (19) and (24) are asymptotically equivalent, that is*

$$\sum_{j=0}^{\infty} \left\| S_{a^j} - S_a \right\|_\infty < \infty.$$

**Proof.** From NSSs (19) and (24), we get

$$\sum_{j=0}^{\infty} \left\| S_{a^j} - S_a \right\|_\infty = \sum_{j=0}^{\infty} \left\{ \left| \mu_0^j - \frac{1}{4} (3 - u_0) \right| + \left| \mu_1^j - \frac{1}{4} (1 + u_0) \right| \right\}.$$

From $(a)$ of Lemma 6, it follows that

$$\sum_{j=0}^{\infty} \left| \mu_1^j - \frac{1}{4} (1 + u_0) \right| \leq \sum_{j=0}^{\infty} C_0 \frac{1}{2^{2j}} < \infty.$$

Similarly, from $(b)$ of Lemma 6,

$$\sum_{j=0}^{\infty} \left| \mu_0^j - \frac{1}{4} (3 - u_0) \right| < \infty.$$

Hence,

$$\sum_{j=0}^{\infty} \left\| S_{a^j} - S_a \right\|_\infty < \infty.$$

□

**Theorem 4.** *The NSS (19) is $C^0$ for $-1 < u_0 < 1$; and $C^1$ for $u_0 = 0$.*

**Proof.** By Lemma 7, $S_a$ is $C^1$ continuity, then it is sufficient to prove that the SS (19) and (24) are asymptotically equivalent:

$$\sum_{j=0}^{\infty} 2^j \left\| S_{a^j} - S_a \right\|_\infty < \infty,$$

where

$$\|S_{a^j} - S_a\|_\infty = max\left\{\sum_{k\in\mathbb{Z}}|a^j_{2k} - a_{2k}, \sum_{j\in\mathbb{Z}}|a^j_{2k+1} - a_{2k+1}|\right\}$$

$$= \sum_{j=0}^{\infty}\left\{2\left|\mu^-_0\frac{1}{4}(3-u_0)\right| + 2\left|\mu^j_1 - \frac{1}{4}(1+u_0)\right|\right\}.$$

Note that

$$\left|\mu^j_0 + \mu^j_1\right| \le \left|\mu^j_0 - \frac{1}{4}(3-u_0)\right| + \left|\mu^j_1 - \frac{1}{4}(1+u_0)\right|.$$

Consider

$$\sum_{j=0}^{\infty}2^j\left|\mu^j_0 - \frac{1}{4}(3-u_0)\right| < \infty \text{ and } \sum_{k=0}^{\infty}2^k\left|\mu^k_1 - \frac{1}{4}(1+u_0)\right| < \infty,$$

by $(a)$ and $(b)$ of Lemma 6, it follows that

$$\sum_{j=0}^{\infty}2^j\left|\mu^j_0 + \mu^j_1 - 1\right| < \infty.$$

Hence,

$$\sum_{j=0}^{\infty}2^j\left\|S_{a^j} - S_a\right\|_\infty < \infty.$$

□

Here, we present the smoothness of four point SS (21). The preceding Lemma can be obtained by using Lemma 6.

**Lemma 9.**

$$(a) \quad \left|\mu^k_{-1} - \left(-\frac{7}{128} + \frac{3u_0}{64}\right)\right| \le D_0\frac{1}{2^{2j}},$$

$$(b) \quad \left|\mu^k_0 - \left(\frac{105}{128} - \frac{5u_0}{64}\right)\right| \le D_1\frac{1}{2^{2j}},$$

$$(c) \quad \left|\mu^k_1 - \left(\frac{35}{128} + \frac{u_0}{64}\right)\right| \le D_2\frac{1}{2^{2j}},$$

$$(d) \quad \left|\mu^k_2 - \left(-\frac{5}{128} + \frac{u_0}{64}\right)\right| \le D_3\frac{1}{2^{2j}},$$

*where $D_0$, $D_1$, $D_2$, and $D_3$ are some constants independent of j.*

**Remark 2.** *Our proposed four point SS (21) is a nonstationary counterpart of the following SS:*

$$\begin{cases} t^{j+1}_{2i} = \left(-\frac{7}{128} + \frac{3u_0}{64}\right)t^j_{i-1} + \left(\frac{105}{128} - \frac{5u_0}{64}\right)t^j_i + \left(\frac{35}{128} + \frac{u_0}{64}\right)t^j_{i+1} + \left(-\frac{5}{128} + \frac{u_0}{64}\right)t^j_{i+2}, \\ t^{j+1}_{2i+1} = \left(-\frac{5}{128} + \frac{u_0}{64}\right)t^j_{i-1} + \left(\frac{35}{128} + \frac{u_0}{64}\right)t^j_i + \left(\frac{105}{128} - \frac{5u_0}{64}\right)t^j_{i+1} + \left(-\frac{7}{128} + \frac{3u_0}{64}\right)t^j_{i+2}, \end{cases} \quad (25)$$

*because the stencils of NSS (21) converge to the stencils of SS (25): $\mu^j_{-1} \to \left(-\frac{7}{128} + \frac{3u_0}{64}\right)$, $\mu^j_o \to \left(\frac{105}{128} - \frac{5u_0}{64}\right)$, $\mu^j_1 \to \left(\frac{35}{128} + \frac{5u_0}{64}\right)$ and $\mu^j_2 \to \left(-\frac{5}{128} + \frac{u_0}{64}\right)$ as $j \to \infty$. The proofs of these convergences follow from Lemma 9.*

- *Using $u_0 = 7/2$ in (21), our four-point NSS becomes a nonstationary counterpart B-spline of degree 6.*
- *Using $u_0 = 0$ in (21), our four-point NSS becomes a nonstationary counterpart of the SSS of [30].*

- Using $u_0 = \frac{19}{6} + 4w$ in (21), our four-point NSS becomes a nonstationary counterpart of the four-point SSS of [24].
- Using $u_0 = \frac{8}{3}$ in (21), our four-point NSS becomes a nonstationary counterpart of the SSS of [31].
- Using $u_0 = 64w + \frac{5}{2}$ in (21), our four-point NSS becomes a nonstationary counterpart of the four-point SSS of [32].

**Lemma 10.** *The Laurent polynomial* $\beta(x)$ *of SS (25) can be written as*

$$
\begin{aligned}
\beta(x) &= \left\{ \left( \frac{u_0}{64} - \frac{5}{128} \right) x^0 + \left( \frac{3u_0}{64} - \frac{7}{128} \right) x^1 + \left( \frac{35}{128} + \frac{u_0}{64} \right) x^2 + \left( \frac{105}{128} - \frac{5u_0}{64} \right) x^3 \right. \\
&\quad \left. + \left( \frac{105}{128} - \frac{5u_0}{64} \right) x^4 + \left( \frac{35}{128} + \frac{u_0}{64} \right) x^5 + \left( \frac{3u_0}{64} - \frac{7}{128} \right) x^6 + \left( \frac{u_0}{64} - \frac{5}{128} \right) x^7 \right\},
\end{aligned}
$$

*and a scheme* $S_\beta$ *corresponding to the Laurent polynomial* $\beta(x)$ *is* $C^4$ *continuity for* $\frac{1}{2} < u_0 < \frac{9}{2}$ *and* $C^5$ *continuity for* $u_0 = \frac{7}{2}$.

**Proof.** Since

$$
b(x) = \frac{32\beta(x)}{(1+x)^5} = \left\{ \left( \frac{-5 + 2u_0}{4} \right) + \left( \frac{18 - 4u_0}{4} \right) x^1 + \left( \frac{-5 + 2u_0}{4} \right) x^2 \right\}.
$$

Note that for $\frac{1}{2} < u_0 < \frac{9}{2}$

$$
\left\| \frac{1}{2} S_b \right\| = \frac{1}{2} \max \left\{ \sum_{k \in \mathbb{Z}} |b_{2k}|, \sum_{k \in \mathbb{Z}} |b_{2k+1}| \right\} = \frac{1}{2} \max \left\{ 2 \left| \frac{-5 + 2u_0}{4} \right|, \left| \frac{18 - 4u_0}{4} \right| \right\} < 1.
$$

Hence, by [30], the SS $S_\beta$ is $C^4$.
To prove $C^5$ smoothness, we put $u_0 = \frac{7}{2}$ in $b(x)$ and we get

$$
b(x) = \frac{1}{2} \left( 1 + 2x + x^2 \right).
$$

If

$$
c(x) = \frac{2b(x)}{(1+x)} = (1 + x),
$$

then, $\left\| \frac{1}{2} S_c \right\| = \frac{1}{2} \max \left\{ \sum_{k \in \mathbb{Z}} |c_{2k}|, \sum_{k \in \mathbb{Z}} |c_{2k+1}| \right\} = \max \left\{ \frac{1}{2}, \frac{1}{2} \right\} < 1.$ Hence, by [30], the SS $S_\beta$ is $C^5$ continuity. $\square$

Now, we prove that the SSs (21) and (25) are asymptotically equivalent.

**Theorem 5.** *The SSs (21) and (25) are asymptotically equivalent, that is*

$$
\sum_{j=0}^{\infty} \left\| S_{a^j} - S_a \right\|_\infty < \infty.
$$

**Proof.** From (21) and (25), we get

$$
\begin{aligned}
\sum_{j=0}^{\infty} \left\| S_{a^j} - S_a \right\|_\infty &= \sum_{j=0}^{\infty} \left\{ \left| \mu_{-1}^j - \left( -\frac{7}{128} + \frac{3u_0}{64} \right) \right| + \left| \mu_0^j - \left( \frac{105}{128} - \frac{5u_0}{64} \right) \right| + \right. \\
&\quad \left. \left| \mu_1^j - \left( \frac{35}{128} + \frac{u_0}{64} \right) \right| + \left| \mu_2^j - \left( -\frac{5}{128} + \frac{u_0}{64} \right) \right| \right\}.
\end{aligned}
$$

From (a) of Lemma 9, it follows that

$$\sum_{j=0}^{\infty} \left| \mu_{-1}^j - \left( -\frac{7}{128} + \frac{3u_0}{64} \right) \right| \leq \sum_{j=0}^{\infty} D_0 \frac{1}{2^{2j}} < \infty.$$

Similarly from (b), (c), and (d) of Lemma 9

$$\left| \mu_0^j - \left( \frac{105}{128} - \frac{5u_0}{64} \right) \right| < \infty, \quad \left| \mu_1^j - \left( \frac{35}{128} + \frac{u_0}{64} \right) \right| < \infty,$$

$$\left| \mu_2^j - \left( -\frac{5}{128} + \frac{u_0}{64} \right) \right| < \infty.$$

Hence,

$$\sum_{j=0}^{\infty} \left\| S_{a^j} - S_a \right\|_\infty < \infty.$$

□

The proof of the following theorem is similar to the proof of Theorem 4.

**Theorem 6.** *The NSS (21) is $C^4$ continuity for $\frac{1}{2} < u_0 < \frac{9}{2}$ and $C^5$ continuity for $u_0 = \frac{7}{2}$.*

## 5. Results and Discussion

In this section, we discuss the geometric comparison of the binary even-point approximating NSSs. In order to present the achievements of the NSSs (19) and (21), we discuss continuity, shape of limit curves, curvature, and torsion. The following general characteristics of proposed SSs are observed from these figures:

- Table 1 depicts the comparison of the smoothness of the proposed ANSSs. It is shown from Table 1 that the proposed ANSSs have higher continuity as compared to other existing NSSs [5,9,10,33–35].
- Figure 1 demonstrates the visual performance of the proposed ANSSs. The limit curves, obtained by our ANSSs, are more rational as compared to other famous existing NSSs. In Figure 2, two examples are given to illustrate the behavior of the parameter $u_0$. The proposed ANSSs gives considerable variations in the results, which is a useful mechanism in geometric modeling.
- In Figure 3, five initial points are sampled to generate limit circle along with curvature. The limit curves generated by NSSs (19) and (21) Reference [5,9] have less tendency to depart from their tangent compared to the limit curves produced by [33–35].
- Figures 4 and 5 indicate that curvature and torsion behavior of limit curves are closely related.

**Table 1.** Comparison of the even-point nonstationary subdivision schemes (NSSs) with existing subdivision schemes (SSs).

| Scheme | Type | $C^n$ | Scheme | Type | $C^n$ | Scheme | Type | $C^n$ |
|---|---|---|---|---|---|---|---|---|
| 2-point [33] | ANSS | $C^1$ | 4-point [11] | INSS | $C^2$ | 4-point [7] | INSS | $C^1$ |
| 2-point [5] | ANSS | $C^1$ | 4-point [35] | INSS | $C^1$ | 4-point [8] | INSS | $C^2$ |
| 3-point [33] | ANSS | $C^3$ | 4-point [9] | INSS | $C^2$ | 2-point (19) | ANSS | $C^1$ |
| 3-point [10] | ANSS | $C^1$ | 6-point [11] | INSS | $C^2$ | 4-point (21) | ANSS | $C^5$ |
| 3-point [5] | ANSS | $C^3$ | 6-point [34] | INSS | $C^2$ | | | |

It is clear from the above comparison that the basic properties of limit curves generated by our proposed NSSs with a smaller number of initial control points have little tendency to depart from their tangents as well as their osculating planes compared to the limit curves obtained by the existing SSSs.

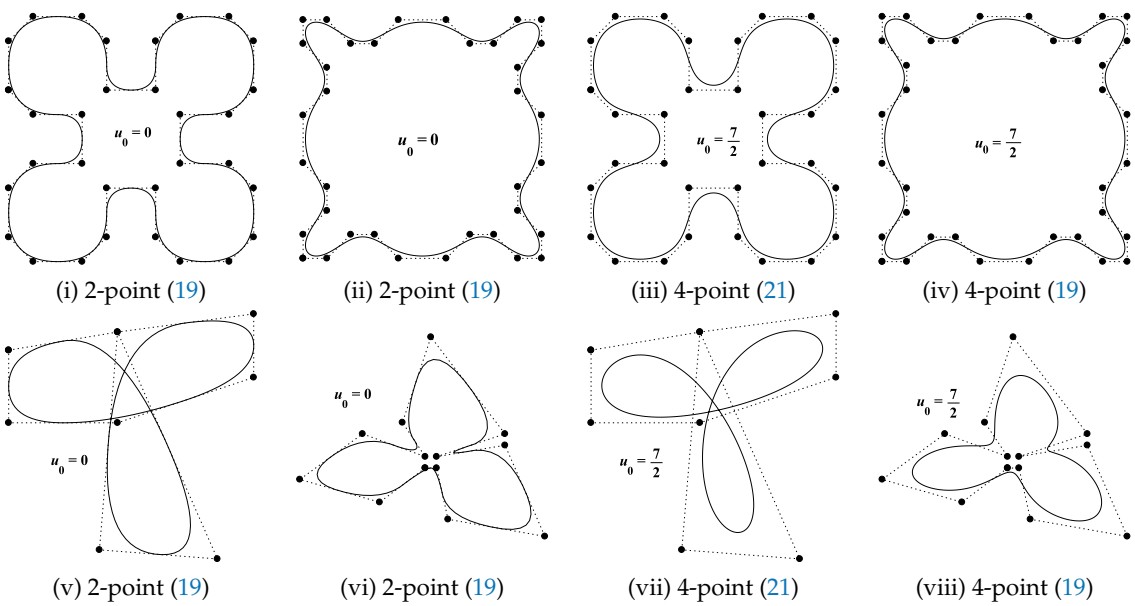

**Figure 1.** Four subdivision levels of SSs (19) and (21) have been used to the control polygon when control points are moved over a short distance, respectively.

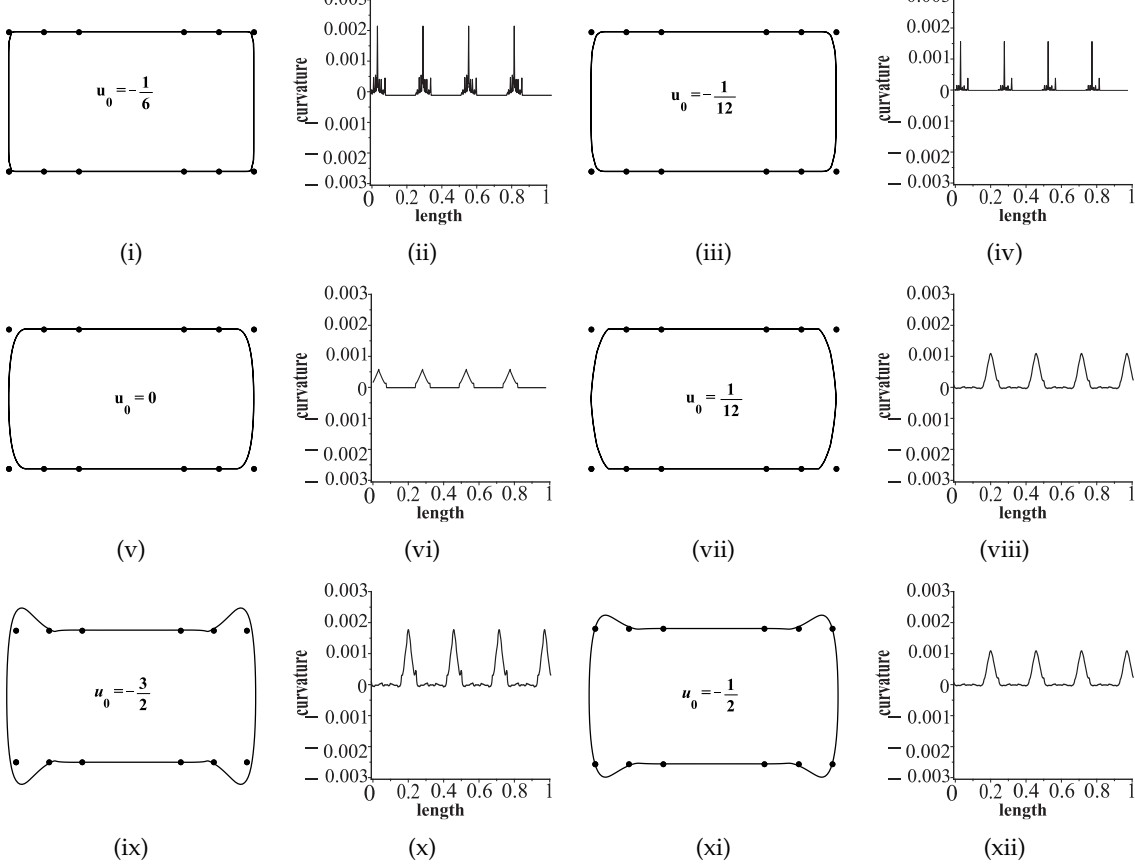

**Figure 2.** *Cont.*

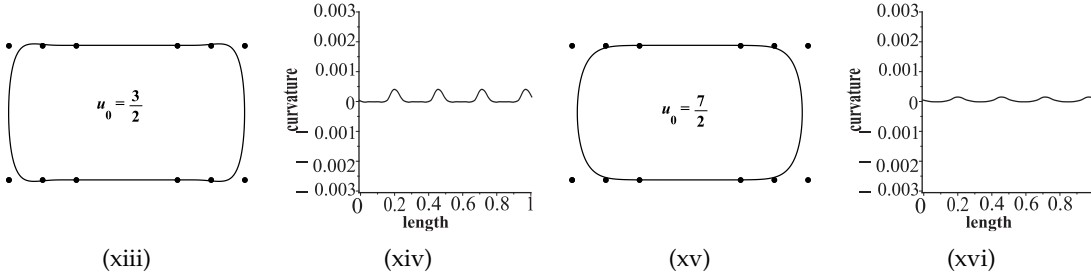

(xiii)       (xiv)       (xv)       (xvi)

**Figure 2.** Four subdivision levels of SS (19) and (21) have been used to the control polygon. The results after different values of $u_0$ are shown on the left together with their corresponding curvature on the right.

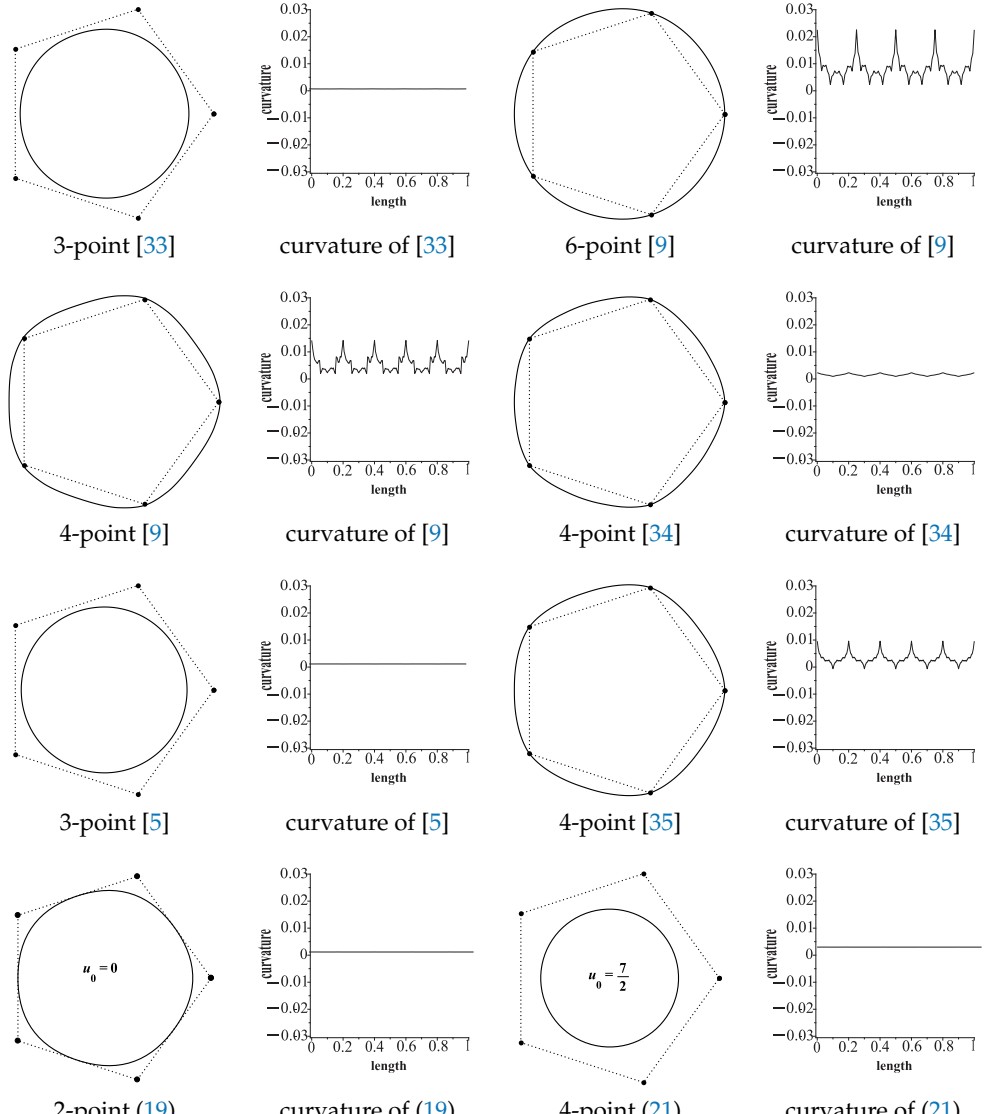

3-point [33]    curvature of [33]    6-point [9]    curvature of [9]

4-point [9]    curvature of [9]    4-point [34]    curvature of [34]

3-point [5]    curvature of [5]    4-point [35]    curvature of [35]

2-point (19)    curvature of (19)    4-point (21)    curvature of (21)

**Figure 3.** Comparison of existing and our proposed SS when five control points are sampled from circle. Limit curves obtained after 5th iteration are shown in the left column, the corresponding curvatures are shown in the right column.

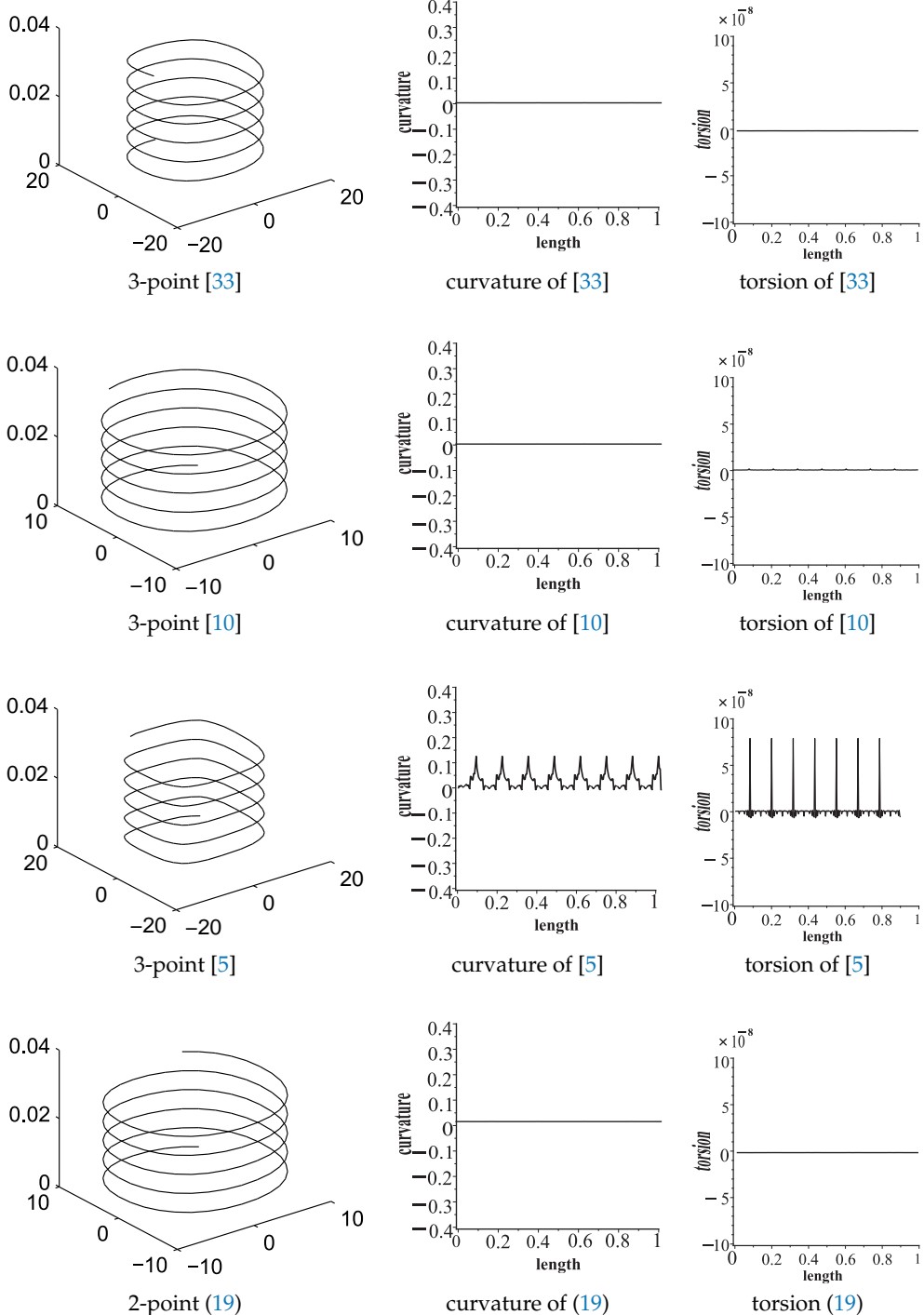

**Figure 4.** Results of 5th iteration of the 3-point schemes [5,10,33] and (19) are shown on the left when the initial control points are sampled from Helix. The corresponding curvatures and torsions are shown in the center and right column, respectively.

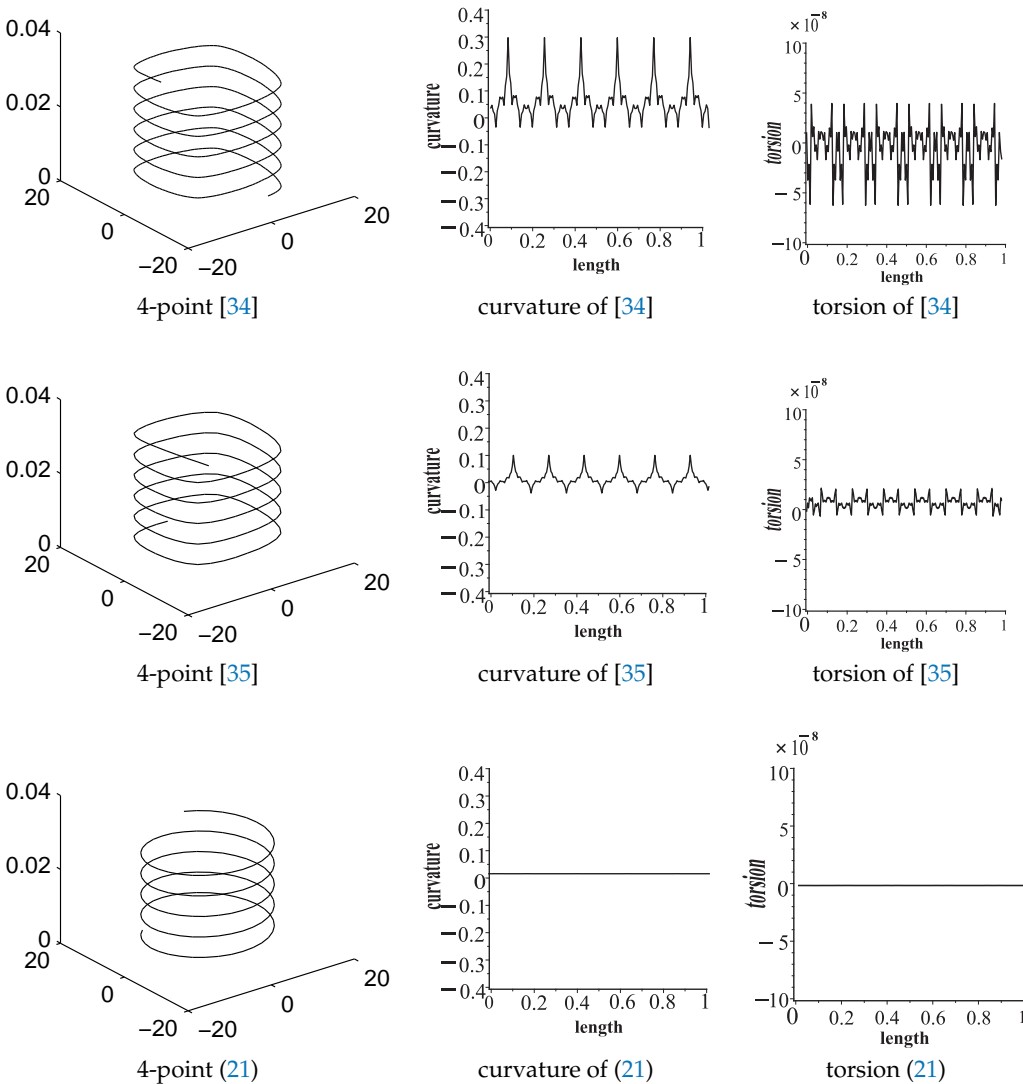

**Figure 5.** Results of 5th iteration of the 4-point and 5-point schemes of [34,35] and (21) are shown on the left when the initial control points are sampled from Helix. The corresponding curvatures and torsions are shown in the center and right column, respectively.

## 6. Conclusions and Future Research

Here, we have established an algorithm for the construction of even-point binary ANSS (for any integer $n \geq 1$) by using Lagrange-like interpolant. It is observed that the newly generated schemes by our proposed algorithm are a nonstationary counterpart of the SSs [2,5,24,30–32,34]. Moreover, in comparison with the schemes that have already emerged [5,9,10,33–35], the newly generated schemes are more refined and better in the sense of smoothness, curvature, and torsion. As observed, the proposed algorithm for different values of $q$ can be considered more universal because it allows us to present general formulae for binary NSSs. These advantages motivate us to extend the proposed results in subdivision surface modeling.

**Author Contributions:** A.G., M.B. and M.I.: Writing original manuscript; Z.U. and K.S.N.: Formal Analysis; K.S.N. and D.B.: Writing review and editing.

**Funding:** This research received no external funding.

**Conflicts of Interest:** The authors declare no conflicts of interest.

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
