# Peer review of "A New Class of 2q-Point Nonstationary Subdivision Schemes and Their Applications"

_mathematics, doi:10.3390/math7070639_

Round 1

Reviewer 1 Report

 I am personally very interested in the topic of te manuscript. But, I think that the manuscript presents several main problems that need to be solved before a referee goes into the analysis of the mathematical details.

First: the English style is very poor: verbs are missing, article are missing, unclear sentences are present and several typos make the understainding more complicated than needed.

Second:  often formulas are not correctly written or not well explained. Several examples follows but other unclear points are present:

- in the mask definition (below formula (1)) what is the variation of the indeces i and  j?

- the use  of the index n in the formulas of Section 2 confuses the reader. Why dont you use \ell, for example?

-formula (2):  what does it mean " almost"  1? It is certainly subjective. Please, explain if condition (1) is an assumption

-why do you evaluate L_n^{2m-1} at 1/4? Is it the method a "dual" method? Provide details

-the masks in (6) depend on m while m is not mentioned. This is uncorrect

-what is k  in formula (7)? What is its variation?

-more important: from where you get the idea of formula (7)? Ca you try to explain it to the reader?

-Is simply U_m=4^{2m-1}n? But then how does n vary?

-In view of the observation above I am not sure that the value of \mu_0^j and \mu_0^j you provide after (8) are correct. I get something different!

-the meaning of the normalized scheme mentioned in Remark 2.1. is not at all obvious. Why do you need a normalization?

-Lemma 3:  are you proving asymptotical similarity?  But, even if based on that notion, convergence of the corresponding basic limit function is something to be proven! Please, have a look to  the paper [C. Conti, N. Dyn, C. Manni, M.-L. Mazure,
Convergence of univariate non-stationary subdivision schemes via asymptotical similarity, Computer Aided Geometric Design
Volume 37 Issue C, (2015), Pages 1--8]

In consideration of the above commets I have stopped the reading  of the manuscript at the end of section 2.

Please, re-write the manuscript more carefully provinding details and stressing the originality of the ideas by using a higher level English style.

Author Response

Response to Reviewer #1

Thank you very much for the reviewer’s valuable comments and suggestions. The comments and responses are as follows:

1)                First: the English style is very poor: verbs are missing, article are missing, unclear sentences are present and several typos make the understanding more complicated than needed.

Response: The English language of the paper is corrected now according to the reviewer’s comments.

2)                Second:  often formulas are not correctly written or not well explained. Several examples follow but other unclear points are present:

Response: Formulas are corrected now according to reviewer’s comments.

3)                - in the mask definition (below formula (1)) what is the variation of the indices i and  j?

Response: in the mask the indices i  and j ?

4)                the use  of the index n in the formulas of Section 2 confuses the reader. Why dont you use \ell, for example?

Response: The use of indexes in Section 2 have been made clear.

a.     -formula (2):  what does it mean " almost"  1? It is certainly subjective. Please, explain if condition (1) is an assumption

b.     -why do you evaluate L_n^{2m-1} at 1/4? Is it the method a "dual" method? Provide details

c.      -the masks in (6) depend on m while m is not mentioned. This is uncorrect

d.     -what is k  in formula (7)? What is its variation? 

e.      -more important: from where you get the idea of formula (7)? Ca you try to explain it to the reader?

f.       -Is simply U_m=4^{2m-1}n? But then how does n vary?

g.     -In view of the observation above I am not sure that the value of \mu_0^j and \mu_0^j you provide after (8) are correct. I get something different!

Response: From equation (2) to (8), we re-write the formulas providing more details and equation numbering is changed.

5)                The meaning of the normalized scheme mentioned in Remark 2.1. is not at all obvious. Why do you need a normalization?

Response: To fulfil the necessary condition and affine invariance property, we need normalization.

6)                Lemma 3:  are you proving asymptotical similarity?  But, even if based on that notion, convergence of the corresponding basic limit function is something to be proven! Please, have a look to the paper [C. Conti, N. Dyn, C. Manni, M.-L. Mazure,
Convergence of univariate non-stationary subdivision schemes via asymptotical similarity, Computer Aided Geometric Design
Volume 37 Issue C, (2015), Pages 1--8]

Response: We have used asymptotically equivalent instead of asymptotical similarity, Please see [17], [18],[20], [30] and [32].

7)                Please, re-write the manuscript more carefully providing details and stressing the originality of the ideas by using a higher level English style.

Response: Essential parts of the manuscript are re-written and language of the paper has been corrected now according to the reviewer’s comments.

Reviewer 2 Report

The paper is interesting and fits the scope of the issue.

Maybe an extension to high dimension will be wellcome.

Author Response

Response to Reviewer #2

Thank you very much for reviewer’s valuable comments and suggestions. The comments and responses are as follows:

Comment: The paper is interesting and fits the scope of the issue. Maybe an extension to high dimension will be welcome.

Response: Thank you very much for the positive comments.

Reviewer 3 Report

The main goal of this paper is the introduction of a new class of non-stationary subdivision algorithms. Subdivision schemes are commonly used in approximation and applications, so, the problem is very important.

The paper is well--written, but the authors should pay attention to articles. Articles (the/a/an) are the main issue.

It should be clearly stated which results are new.

Author Response

Response to Reviewer #3

Thank you very much for reviewer’s valuable comments and suggestions. The comments and responses are as follows:

Comment: The paper is well--written, but the authors should pay attention to articles. Articles (the/a/an) are the main issue.

Response: Articles (the/a/an) are corrected now according to reviewer’s comments.

Comment: It should be clearly stated which results are new.

Response: The main results mentioned in this paper are newly generated. At the end of the introduction section, the contribution of the paper are mentioned clearly.

Round 2

Reviewer 1 Report

I The revised version of the manuscript is better but, to be honest, I am not yet satisfied by the revision. The feeling I have is that the revision was made in a hurry (actually i was surprise to get already the revision). As an example just consider the description of the plan of the article at page 2:
 Section 1 rather than Section 2 is devoted to basic identities and definitions, smoothness and convergence is discussed in Section 2 and not in Section 3.  Sections 4 and 5 do not correspond to what you write! Conclusion are drown in Section 4 not in 5!

 Some critical points have been fixed but many other criticisms now appear: what is the meaning Remark 1? The presentation of the symbol of a scheme? The symbol is the z-transfrom of the mask and the variable should be z to be in agrement with the literature. It is not clear where the Remark ends and where a new part starts.

Also,  look at line 79 of page 4 that refers to (11) not yet introduced. How can it be?

  All the presentation of the first part of the manuscript is very confusing and misleading. I did not understand the flow of the presentation.  Please, take your time to re-write the paper carefully also thinking to its organisation, providing a presentation that a reader can follow.  For example, in Section 1, I would separate the preliminary technical results form the new material and I would present the general scheme (15) along with an explanation of the formulas? Why do you use (16)? I mean from where it comes?

 Moreover, even if the scheme in (15) is given for general q Section 2 you actually considers (18) and (20) only. Can you provide any result for general q?

In Section 3 you mention the comparison of your schemes with the result obtained by existing SSSs without providing any picture of the latter! This is not good since and interested reader is supposed to search for the other type of results corresponding to your input data and this is not an easy task.
Moreover, if I understood this section  correctly,  shape of the limit curves, curvature and torsion are just discussed numerically. If it is so, this should be clearly stated in the introduction. What you write now in the introduction it is not clear at all.  But, certainly a more theoretical  approach would be preferable.

Author Response

Response to the reviewer

The authors are very grateful to the anonymous referees for giving valuable comments and suggestions, which help us to improve the paper in the current form. We have revised manuscript according to the referee’s suggestions. Details are given below:

Comments and Response to the Reviewer

1)      The revised version of the manuscript is better but, to be honest, I am not yet satisfied by the revision. The feeling I have is that the revision was made in a hurry (actually I was surprise to get already the revision). As an example just consider the description of the plan of the article at page 2:

2)       Section 1 rather than Section 2 is devoted to basic identities and definitions, smoothness and convergence is discussed in Section 2 and not in Section 3.  Sections 4 and 5 do not correspond to what you write! Conclusion are drown in Section 4 not in 5?

Response: The organization of the paper is corrected now according to the reviewer’s comments.

3)      Some critical points have been fixed but many other criticisms now appear: what is the meaning Remark 1? The presentation of the symbol of a scheme? The symbol is the z-transform of the mask and the variable should be z to be in agreement with the literature. It is not clear where the Remark ends and where a new part starts.:

Response: Remark 1 is removed and in the z-transform of the mask,  the variable z has added according to the reviewer’s comments.

4)      Also, look at line 79 of page 4 that refers to (11) not yet introduced. How can it be?

Response: It is corrected.

5)      All the presentation of the first part of the manuscript is very confusing and misleading. I did not understand the flow of the presentation.  Please, take your time to re-write the paper carefully also thinking to its organization, providing a presentation that a reader can follow.  For example, in Section 1, I would separate the preliminary technical results from the new material and I would present the general scheme (15) along with an explanation of the formulas? Why do you use (16)?

Response: We have re-written the paper carefully and separated the preliminary section. Now the presentation of the first part of the manuscript is very clear.

6)      - Moreover, even if the scheme in (15) is given for general q Section 2 you actually considers (18) and (20) only. Can you provide any result for general q?

Response: Yes the scheme in (15) (Now changed to 17) is given for general q in Section 2. To avoided the complexity we have generated the mask of scheme for q=1,2. From this formula, anyone can get other even point schemes.

7)      - In Section 3 you mention the comparison of your schemes with the result obtained by existing SSSs without providing any picture of the latter! This is not good since and interested reader is supposed to search for the other type of results corresponding to your input data and this is not an easy task.

Moreover, if I understood this section correctly, shape of the limit curves, curvature and torsion are just discussed numerically. If it is so, this should be clearly stated in the introduction. What you write now in the introduction it is not clear at all.  But, certainly a more theoretical approach would be preferable.

Response: In Figure 3, 4 and 5,   we have compared the limit curve, curvature, and torsion of the proposed scheme to the existing schemes.

  Also, we clearly stated about the shape of the limit curves, curvature and torsion in the introduction according to reviewer’s comments.
